# Jumps and Cojumps analyses of major and minor cryptocurrencies

**Piyachart Phiromswad[1], Pattanaporn Chatjuthamard[1], Sirimon Treepongkaruna[1,2], Sabin Srivannaboon** [1]*

1 Center of Excellence in Management Research for Corporate Governance and Behavioral Finance, Sasin School of Management, Chulalongkorn University, Bangkok, Thailand, 2 UWA Business School, The University of Western Australia, Crawley, WA, Australia

* sabin.srivannaboon@sasin.edu

## Abstract

This paper empirically examines jumps and cojumps of both major and minor cryptocurrencies. Understanding the nature of their jumps and cojumps plays an important role in risk management, asset allocation and pricing of derivatives. We find that all cryptocurrencies display significant jumps. Furthermore, minor cryptocurrencies appear to have significantly higher jump intensity and jump size than major cryptocurrencies. Finally, we find that cojumps of the Thai stock market index and minor cryptocurrencies have a greater intensity than that of major cryptocurrencies.

## Introduction

Evidences of jumps and cojumps are abundant. Studies found jumps in stock prices [1–4], currencies [4–6], bond prices [4, 7], interest rates [8] and even in electricity prices [9]. Cojumps are also found in stock prices [3, 10] and exchange rates [5]. Furthermore, researchers have found that jumps and cojumps are likely to be driven by company-specific and overall market-level news. Evidences can be found in Barndorff-Nielsen and Shephard [11], Bollerslev et al. [10], Lee and Mykland [12], Lahaye et al. [13], Gilder et al. [3] and Chatrath et al [5]. However, only limited studies have examined the presence of jumps and cojumps in cryptocurrencies. The total market capitalization of all cryptocurrencies in May 2020 was 250 Billion USD which is approximately half of the GDP of Thailand and this number is growing fast. Among the cryptocurrencies, even less research has examined the presence of jumps and cojumps in the minor cryptocurrencies. Typically, researchers focus on popular cryptocurrencies such as Bitcoin, Ethereum and Litecoin. However, the total market capitalization of the 11th to the 500th cryptocurrencies ranked by the market capitalization is 41 Billion USD which is about 70 percent of the market capitalization of the top 10 cryptocurrencies by market capitalization (excluding Bitcoin). Therefore, examining the presence of jumps and cojumps in minor cryptocurrencies remains to be a major literature gap.

Understanding the nature of jumps and cojumps of the major as well as the minor cryptocurrencies will be important for risk management, asset allocation and pricing of derivatives. For example, cryptocurrencies can be used for hedging against other assets. Hedging capabilities can be examined by investigating correlation properties of assets (or asset classes) via

**Data Availability Statement:** There are two main datasets for this paper. The data could not be uploaded because the data are owned by a third-party organization. But it is possible to purchase and/or obtain the data to replicate the results of this

**Funding:** Funding Information The research was funded by Chulalongkorn University under the Ratchadapisek Sompoch Endowment Fund (2020) through Collaborating Centre for Labor Research at Chulalongkorn University (CU-Collar) (763008) and Center of Excellence in Management Research for Corporate Governance and Behavioral Finance. The funders had no role in study design, data collection and analysis, decision to publish, or preparation of the manuscript.

**Competing interests:** The authors have declared that no competing interests exist.

constant conditional correlation (CCC) model or dynamic conditional correlation (DCC) model (Bollerslev [14]; Engle [15]). Dyhrberg [16] examined the hedging capabilities of Bitcoin in reference to gold and the US Dollar. She found that Bitcoin can be used for risk management especially when investors anticipate negative news. Baur et al. [17] examined the return properties of Bitcoin and found that it is quite different from several typical assets both in normal periods and in crisis periods. This implies that Bitcoin can be used to form a portfolio that can achieve greater diversification benefits. Nevertheless, extremely limited research in these topics has been done for minor cryptocurrencies.

Among a few studies that examine jumps and cojumps in cryptocurrencies are Bouri et al. [18] who examined jumps and cojumps of 12 cryptocurrencies (Bitcoin, Bitshares, Bytecoin, Dash, Digibyte, Dogecoin, Ethereum, Litecoin, Monero, Nem, Ripple and Stellar). They found that these cryptocurrencies exhibit significant jumps except for Ripple, Bitcoin and Litecoin. Other examples are Chaim and Laurini [19] and Scaillet et al. [20] who examined and found the presence of jumps in Bitcoin. Indirectly, Cheah et al. [21]. Examined the interdependency of cross-market bitcoin prices and found that they are cointegrated. They argued that this is a sign of market inefficiency in bitcoin markets in which investors can speculate for profits. Katsiampa [22]. Examines interdependency in the two cryptocurrencies which are Bitcoin and Ethereum. She found that there are interdependencies between the two cryptocurrencies as well as some evidence indicating that major news influences the correlation and volatility of these assets. Chaim and Laurini [23] examined the joint dynamic of nine major cryptocurrencies. They assumed in their model that jumps to return and volatility are common among the nine cryptocurrencies. Their model allows the distinction between permanent and transitory jumps in which they found to be presence and increasing more frequent starting from early 2017.

In this paper, we contribute to the existing literature by empirically examining jumps and cojumps of both major and minor cryptocurrencies. In total, we examine 54 cryptocurrencies. We also empirically examine cojumps of these cryptocurrencies with the Thai stock market. To the best of our knowledge, this is the most extensive study of jumps and cojumps of cryptocurrencies in the literature. S1 Table presents the list of cryptocurrencies (symbols) considered in this paper as well as their full name and the associated market capitalization according to CoinMarketCap (accessed on June, 2020). Among these cryptocurrencies, 3 cryptocurrencies have market capitalization more than 10 Billion USD, 10 cryptocurrencies have market capitalization between than 1 Billion USD and 10 Billion USD, 23 cryptocurrencies have market capitalization between than 100 Million USD and 1 Billion USD and 18 cryptocurrencies have market capitalization less than 100 Million USD. A related paper that examines a extensive set of cryptocurrencies is Bariviera [24] who examined 84 cryptocurrencies. However, he did not focus on detecting jumps and cojumps as we did in this paper. Our methodology in detecting jumps is based on Barndorff-Nielsen and Shephard's [25] bi-power variation. We find that all cryptocurrencies display significant jumps. Furthermore, minor cryptocurrencies appear to have significantly higher jump intensity and jump size than major cryptocurrencies do. Finally, we find that cojumps of the Thai stock market index and minor cryptocurrencies have a greater intensity than that of major cryptocurrencies. These findings are novel and have not been documented in the literature.

## Methodology

Following Andersen and Bollerslev [26] and Barndorff-Nielsen and Shephard [27], let the realized variation ($RV$) be defined as follows:

$$RV_t(\theta) = \sum_{j=1}^{1/\theta} r_{t+j\theta,\theta}^2, t = 1, \ldots, T \tag{1}$$

where $r_{t,\theta} \equiv p(t) - p(t-\theta)$ is defined as the high-frequency intraday returns, $\theta$ is the high-frequency interval (5 minutes in this paper) to measure the intraday returns and $1/\theta$ is the number of 5-minute intervals within a trading day. As $\theta \to \infty$, the realized variation will converge in probability to the continuous sample path variation ($CV$) and the jump variation ($JV$).

Barndorff-Nielsen and Shephard [25] proposed a nonparametric procedure that use the bi-power variation measure ($BV$) to estimate the jump variation. Let the bi-power variation measure (BV) be defined as follows:

$$BV_t(\theta) \equiv \mu_1^{-2} \sum_{j=2}^{1/\theta} |r_{t+j\theta,\theta}^2||r_{t+(j-1)\theta,\theta}^2| \qquad (2)$$

where $\mu_1 \equiv \sqrt{2/\pi}$ is the scaling factor. Barndorff-Nielsen and Shephard [25] provided a proof that the bi-power variation in Eq (2) converges in probability to the continuous sample path variation as follows:

$$BV_t(\theta) \underset{p}{\to} \int_{t-1}^{t} \sigma^2(s)ds \qquad (3)$$

Therefore, the jump variation can be estimated as the difference between the realized variation and the bi-power variation (i.e. $JV_t = RV_t - BV_t$).

Furthermore, we follow the standard practice in the literature by considering only significant jumps. Jumps will be considered significant if its Z-statistic, $Z_t(\theta)$, is greater than a predefined critical value $\Phi_{1-\alpha}$ where $\alpha$ denotes the significance level. We use the Z-statistic proposed by Huang and Tauchen [28]. With this information, continuous variation path $CV_t(\theta)$ is simply the residual of the realized variation from the jump variation. Finally, we determine the sign of the jump using the sign of the largest absolute intraday return as follows:

$$sgn_t(\theta) = Ind(\max_{j=1,\dots,1/\theta} |r_{t,j}|) \qquad (4)$$

where the sign indicator $Ind(.)$ is equal to 1 or –1 depending upon the sign of the largest absolute intraday return.

The statistics that we are interested for jumps analysis are as follows. The jump intensity ($\lambda$) which is the proportion of trading days with significant jumps. The jump intensity is a statistic that can be used to determine whether jumps are prevalence in the testing assets or not. Typically, the jump intensity ($\lambda$) greater than 10 percent has been used a minimum threshold for the prevalence of jumps in stock markets. The jump mean ($\gamma$) and the jump standard deviation ($\delta$) is simply the average and the standard deviation of the (square rooted) signed jump variation (i.e. the average and the standard deviation of $sqn_t(\theta)\sqrt{JV_{t,\alpha}(\theta)}$) for $t = 1,\dots,T$) for days with significant jumps. The jump mean ($\gamma$) will allow us to compare the typical magnitude of jumps across the testing assets while the jump standard deviation ($\delta$) will allow us to compare the variability of jumps across the testing assets. As the (square rooted) signed jump variation for days with significant jumps can be positive and negative, the jump mean could be close to zero as components are cancelling out with one another. Therefore, we also consider the mean and the standard deviation of the absolute jump variation (i.e. (i.e. the average and the standard deviation of $|sqn_t(\theta)\sqrt{JV_{t,\alpha}(\theta)}|$) for $t = 1,\dots,T$) on days with significant jumps ($\gamma^*$ and $\delta^*$ by order). Finally, the jump contribution which captures the relative contribution of jumps in driving the dynamic of return series is defined as the following ratio:

$$jump\ contribution = (\sum_{t=1}^{T} \sqrt{JV_{t,\alpha}(\theta)})/(\sum_{t=1}^{T} \sqrt{RV_{t,\alpha}(\theta)}) \qquad (5)$$

For detecting cojumps, we define cojumps as simultaneous significant jumps between two or more assets on trading days. In this paper, we consider bivariate cojumps where the SET100 index is used as a reference asset for detecting cojumps between each cryptocurrency and Thai stock market. We are interested to learn whether there are common components that move both the cryptocurrencies and the Thai stock market. The statistics that we are interested for cojumps analysis are as follows. Cojump intensity ($\phi$) is defined as the proportion of trading days with simultaneous (significant) jumps between a pair of testing asset. Positive cojump intensity ($\phi_p$) and negative cojump intensity ($\phi_n$) are defined in a similar manner as the cojump intensity ($\phi$) except that the positive cojump intensity counts for days that jumps of both assets are (signed as) positive and the negative cojump intensity counts for days that jumps of both assets are (signed as) negative. Finally, two types of opposed cojump intensity are considered. The positive opposed cojump intensity ($\beta_p$) counts for days that jumps of the testing cryptocurrency is (signed as) positive while the SET100 index is negative. The negative opposed cojump intensity ($\beta_p$) counts for days that jumps of the testing cryptocurrency is (signed as) negative while the SET100 index is positive.

## Data

We follow Andersen, Bollerslev and Das [29] and choose 5-minute interval for measuring the high-frequency intraday returns. Andersen, Bollerslev and Das [29] provided a justification that 5-minute interval is optimal for addressing microstructure biases such as bid-ask bounce, price discreteness and nonsynchronous trading. Our sample period is from March 2, 2018 to August 31, 2018. The intraday data for the SET100 index is from the Stock Exchange of Thailand. As there are more than 2,000 cryptocurrencies in the world (website CoinMarketCap reported 2,085 cryptocurrencies with market capitalization information as of July 2020), it is not possible to study all of them due to data unavailability. In total, only 54 cryptocurrencies have complete data over our sample period of March 2, 2018 to August 31, 2018. We obtain intraday data for cryptocurrencies from the Binance. We use the Python programming language to extract data from the Binance website. A major advantage of this website is its web API which allows us to retrieve historical data for the period that we want and also in a data structure of our choice. S1 Table presents the list of cryptocurrencies (symbols) considered in this paper as well as their full name and the associated market capitalization according to Coin-MarketCap (accessed on June, 2020).

For all of our analyses, we exclude weekend and holidays in Thailand to properly examine cojumps of the Thai stock market and cryptocurrencies. All cryptocurrencies are denominated in BTC (Bitcoin), except for Bitcoin in USD. Cryptocurrencies are divided into four groups based on their market capitalization as follows: (i) cryptocurrencies with market capitalization more than 10 Billion USD (3 cryptocurrencies), (ii) cryptocurrencies with market capitalization between than 1 Billion USD (10 cryptocurrencies) and 10 Billion USD, (iii) cryptocurrencies with market capitalization between than 100 Million USD and 1 Billion USD (23 cryptocurrencies) and cryptocurrencies have market capitalization less than 100 Million USD (18 cryptocurrencies). Cryptocurrencies with market capitalization more than 1 Billion USD are considered as major cryptocurrencies while the rest are considered as minor cryptocurrencies.

The total market capitalization of the top 10 cryptocurrencies (excluding Bitcoin which has a market capitalization of 168 Billion USD) is 59 Billion. The total market capitalization of the 11th to the 500th cryptocurrencies ranked by the market capitalization is 41 Billion USD which is about 70 percent the market capitalization of the major cryptocurrencies (excluding Bitcoin). Therefore, the minor cryptocurrencies could play an important in financial markets and should receive more attention.

## Empirical findings

### The distribution of Realized Variation (RV), continuous sample path variation (CV) and Jump Variation (JV)

Panel A of Table 1 indicates that the mean and standard deviation of the realized variation of cryptocurrencies (both major and minor) are significantly higher than the SET100 index. The mean and standard deviation of the realized variation of SET100 index 0.0073 and 0.0019 by order (reported in S2 Table) while the means and standard deviations of the major and minor cryptocurrencies are several times higher these values. This is consistent with the notion that cryptocurrencies are typically more suitable for risk seeking investors [16]. For the third (skewness) and the fourth (kurtosis) moments, we find that all series are slightly right-skewed but their kurtosis are quite different from one another. Comparing the major and minor cryptocurrencies, the minor ones are significantly more fat-tailed than the major ones do (the average excess kurtosis of minor cryptocurrencies is 18.62 while the average excess kurtosis of major cryptocurrencies is only 4.06). This gives some indication that there could be more significant jumps in the minor cryptocurrencies relative to the major cryptocurrencies. Panels B and C of Table 1 illustrate the four moments of the distribution of the continuous path

Table 1. Summary statistics for realised variation, continuous sample path variation and jump variation.

| | Major Cryptocurrencies | | | Minor Cryptocurrencies | | |
|---|---|---|---|---|---|---|
| | Market capitalization more than 10 Billion USD (3 Cryptocurrencies) | Market capitalization between than 1 Billion USD and 10 Billion USD (10 Cryptocurrencies) | Average | Market capitalization between than 100 Million USD and 1 Billion USD (23 Cryptocurrencies) | Market capitalization less than 100 Million USD (18 Cryptocurrencies) | Average |
| Panel A: Realised Variation | | | | | | |
| Mean | 0.0330 | 0.0500 | 0.0415 | 0.0664 | 0.0739 | 0.0701 |
| Std | 0.0137 | 0.0200 | 0.0169 | 0.0322 | 0.0317 | 0.0320 |
| Kurtosis | 2.0423 | 6.0927 | 4.0675 | 19.2142 | 18.0327 | 18.6234 |
| Skewness | 1.3491 | 1.8536 | 1.6014 | 3.3241 | 3.1224 | 3.2233 |
| Obs (days) | 123 | 123 | 123 | 123 | 123 | 123 |
| Panel B: Continuous Sample Path Variation | | | | | | |
| Mean | 0.0322 | 0.0487 | 0.0404 | 0.0638 | 0.0705 | 0.0671 |
| Std | 0.0137 | 0.0197 | 0.0167 | 0.0313 | 0.0296 | 0.0304 |
| Kurtosis | 2.0945 | 5.5752 | 3.8349 | 16.3493 | 15.1418 | 15.7455 |
| Skewness | 1.3837 | 1.7926 | 1.5881 | 3.0411 | 2.8747 | 2.9579 |
| Obs (days) | 123 | 123 | 123 | 123 | 123 | 123 |
| Panel C: Jump Variation | | | | | | |
| Mean | 0.0033 | 0.0057 | 0.0045 | 0.0099 | 0.0132 | 0.0116 |
| Std | 0.0063 | 0.0104 | 0.0083 | 0.0158 | 0.0205 | 0.0182 |
| Kurtosis | 2.7346 | 5.9779 | 4.3563 | 12.1436 | 11.9933 | 12.0685 |
| Skewness | 1.9131 | 2.1299 | 2.0215 | 2.5625 | 2.3210 | 2.4418 |
| Obs (days) | 123 | 123 | 123 | 123 | 123 | 123 |

This table presents statistics that summarize the unconditional distributions of daily (square rooted) realized variation, continuous sample path variation and jump variation of the major cryptocurrencies and the minor cryptocurrencies. Panel A is the unconditional distributions of daily realized variation. Panel B is the unconditional distributions of daily continuous sample path variation. Panel C is the unconditional distributions of daily jump variation. S1 Table presents the list of cryptocurrencies (symbols) considered in this paper as well as their full name and the associated market capitalization according to CoinMarketCap (accessed on June, 2020).

variation and jump variation by order. Interestingly, all the above observations remain intact for the continuous path variation and jump variation. S2 to S4 Tables report detailed information about the realized variation, continuous path variation and jump variation of all cryptocurrencies considered in this paper.

## The existence of jumps and cojumps

Panel A of Table 2 reports the existence of jumps. Examining the jump intensity ($\lambda$), we find a higher percentage of days with significant jumps in minor cryptocurrencies (36.21 percent), compared to that in major cryptocurrencies (24.50 percent). The finding that minor cryptocurrencies display significantly greater jump intensity than major cryptocurrencies is novel and has not been documented in the literature.

The signed jump mean ($\gamma$) and the signed jump standard deviation ($\delta$) of all series are relatively small. However, we find that the absolute jump mean ($\gamma^*$) and the absolute jump standard deviation ($\delta^*$) are considerably larger indicating that the positive and the negative jumps

**Table 2. Jumps and Cojumps parameter estimates.**

| | Major Cryptocurrencies | | | Minor Cryptocurrencies | | |
|---|---|---|---|---|---|---|
| | Market capitalization more than 10 Billion USD (3 Cryptocurrencies) | Market capitalization between than 1 Billion USD and 10 Billion USD (10 Cryptocurrencies) | Average | Market capitalization between than 100 Million USD and 1 Billion USD (23 Cryptocurrencies) | Market capitalization less than 100 Million USD (18 Cryptocurrencies) | Average |
| | Panel A: Jumps Parameter Estimates | | | | | |
| $\lambda$ | 0.2249 | 0.2650 | 0.2450 | 0.3340 | 0.3902 | 0.3621 |
| $\gamma^*$ | 0.0033 | 0.0057 | 0.0045 | 0.0099 | 0.0132 | 0.0116 |
| $\delta^*$ | 0.0063 | 0.0104 | 0.0083 | 0.0158 | 0.0205 | 0.0182 |
| $\gamma$ | 0.0001 | 0.0008 | 0.0005 | 0.0005 | 0.0011 | 0.0008 |
| $\delta$ | 0.0071 | 0.0119 | 0.0095 | 0.0188 | 0.0245 | 0.0216 |
| Jump Con | 0.0976 | 0.1122 | 0.1049 | 0.1471 | 0.1775 | 0.1623 |
| Obs (days) | 123 | 123 | 123 | 123 | 123 | 123 |
| | Panel B: Cojumps Parameter Estimates | | | | | |
| $\phi$ | 0.0271 | 0.0341 | 0.0306 | 0.0520 | 0.0583 | 0.0551 |
| $\phi_P$ | 0.0054 | 0.0049 | 0.0051 | 0.0113 | 0.0104 | 0.0108 |
| $\phi_n$ | 0.0081 | 0.0114 | 0.0098 | 0.0194 | 0.0221 | 0.0208 |
| $\beta_n$ | 0.0108 | 0.0106 | 0.0107 | 0.0138 | 0.0163 | 0.0150 |
| $\beta_P$ | 0.0027 | 0.0073 | 0.0050 | 0.0074 | 0.0095 | 0.0085 |
| Obs (days) | 123 | 123 | 123 | 123 | 123 | 123 |

Panel A presents parameter estimates of jump intensity ($\lambda$) which is the proportion of trading days with significant jumps, absolute (or unsigned) jump mean ($\gamma^*$), absolute (or unsigned) jump standard deviation ($\delta^*$), (signed) jump mean ($\gamma$) which is the mean of the (square rooted) jump variation, (signed) jump standard deviation ($\delta$) which is the standard deviation of the (square rooted) jump variation, and jump contribution $(\sum_{t=1}^{T}\sqrt{JV_{t,\alpha}(\theta)})/(\sum_{t=1}^{T}\sqrt{RV_{t,\alpha}(\theta)})$ for the major cryptocurrencies and the minor cryptocurrencies. Panel B reports parameter estimates of cojump intensity ($\phi$) which is defined as the proportion of trading days with simultaneous (significant) jumps between a pair of testing currency with respect to SET100 index, positive cojump intensity ($\phi_p$) which is the proportion of days that jumps of both assets are (signed as) positive, negative cojump intensity ($\phi_n$) which is the proportion of days that jumps of both assets are (signed as) negative, opposed jumps where currency jumps are negative ($\beta_p$), and opposed jumps where currency jumps are positive ($\beta_n$) for the major cryptocurrencies and the minor cryptocurrencies. S1 Table presents the list of cryptocurrencies (symbols) considered in this paper as well as their full name and the associated market capitalization according to CoinMarketCap (accessed on June, 2020).

are equally presence and cancelling each other out. Furthermore, we document that the absolute jump mean ($\gamma^*$) of the minor cryptocurrencies is three times larger than the absolute jump mean ($\gamma^*$) of the major cryptocurrencies (0.0116 as compared to 0.0045). Therefore, not only jumps among the minor cryptocurrencies are more frequency, they are considerably larger in size as well. This result is consistent with the findings of Bariviera [24]. Using multifractal analysis techniques, he found that cryptocurrencies of different trading volume have different stochastic properties. In particular, the cryptocurrencies with large trading volume (i.e. large market capitalization in our context) have monofractal processes while the cryptocurrencies with small trading volume (i.e. small market capitalization in our context) have multifractal processes.

Panel B of Table 2 examines the existence of cojumps and their relative importance. We find that the cojump intensity ($\phi$) of the minor cryptocurrencies (5.5 percent) is twice as large as the cojump intensity ($\phi$) of the major cryptocurrencies (3 percent). Finally, the positive cojump ($\phi_p$), negative cojump ($\phi_n$) and two opposing cojumps ($\beta_p$ and $\beta_n$) indicators are evenly distributed for all series. S5 and S6 Tables report detailed information about estimated jumps and cojumps parameters of all cryptocurrencies considered in this paper.

## Conclusion

Evidences of jumps and cojumps are abundant. Researchers have found evidences of jumps and cojumps in stock prices, currencies, interest rates and even in electricity prices. However, very few studies have examined the presence of jumps and cojumps in cryptocurrencies which will become a major class of asset in the near future. As of May 2020, the total market capitalization of all cryptocurrencies is 250 Billion U.S. dollars, approximately half of the GDP of Thailand and this number is growing fast.

Among existing studies in cryptocurrencies, even fewer papers have examined the presence of jumps and cojumps in the minor cryptocurrencies. Typically, researchers focus on popular cryptocurrencies such as Bitcoin, Ethereum and Litecoin. However, the total market capitalization of the 11th to the 500th cryptocurrencies ranked by the market capitalization is 41 Billion USD, which is about 70 percent of the market capitalization of the top 10 cryptocurrencies (excluding Bitcoin). Therefore, understanding the nature of jumps and cojumps of minor cryptocurrencies plays an important role in risk management, asset allocation and pricing of derivatives as well.

Our paper is the first to examine jumps and cojumps of cryptocurrencies and includes minor cryptocurrencies. In total, we cover 54 cryptocurrencies. To the best of our knowledge, this is the most extensive study of jumps and cojumps of cryptocurrencies in the literature. In detecting jumps, we adopt technique based on Barndorff-Nielsen and Shepherd's [25] bipower variation. Overall. we find all cryptocurrencies display significant jumps. Furthermore, minor cryptocurrencies appear to have higher jump intensity (two times bigger), higher jump size (three times bigger) and higher cojump intensity (two times bigger) with the SET100 index than major cryptocurrencies do. These findings are novel and have not been documented in the literature.

All the above findings, together with the fact that the total market capitalization of the minor cryptocurrencies is large, clearly imply that the minor cryptocurrencies could play an important in financial markets and should receive more attention among researchers. Thus, the minor cryptocurrencies will play an important role for risk management, asset allocation and pricing of derivatives. Regulators should not disregard the minor cryptocurrencies in formulating policies to govern the volatility induced by cryptocurrencies. Similarly, investors should not disregard the minor cryptocurrencies in their risk management and

asset allocation. Even individual minor cryptocurrencies are small in their market capitalization, but as a group, they are large and more volatile than the major cryptocurrencies do.

## Supporting information

**S1 Table. List of cryptocurrencies.**
(DOCX)

**S2 Table. Summary statistics for realised variation.**
(DOCX)

**S3 Table. Summary statistics for continuous sample path variation.**
(DOCX)

**S4 Table. Summary statistics for jump variation.**
(DOCX)

**S5 Table. Jumps parameter estimates.**
(DOCX)

**S6 Table. Cojumps parameter estimates.**
(DOCX)

## Author Contributions

**Conceptualization:** Piyachart Phiromswad, Pattanaporn Chatjuthamard, Sirimon Treepongkaruna, Sabin Srivannaboon.

**Formal analysis:** Piyachart Phiromswad, Pattanaporn Chatjuthamard, Sirimon Treepongkaruna, Sabin Srivannaboon.

**Investigation:** Piyachart Phiromswad, Pattanaporn Chatjuthamard, Sirimon Treepongkaruna, Sabin Srivannaboon.

**Methodology:** Piyachart Phiromswad, Pattanaporn Chatjuthamard, Sirimon Treepongkaruna, Sabin Srivannaboon.

**Project administration:** Sabin Srivannaboon.

**Validation:** Piyachart Phiromswad, Sirimon Treepongkaruna.

**Writing – original draft:** Piyachart Phiromswad, Pattanaporn Chatjuthamard, Sirimon Treepongkaruna, Sabin Srivannaboon.

**Writing – review & editing:** Piyachart Phiromswad, Pattanaporn Chatjuthamard, Sirimon Treepongkaruna, Sabin Srivannaboon.

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
