## [Decision Letter · Decision Letter 0]

14 Oct 2020

PONE-D-20-30502

Jumps and Cojumps Analyses of Major and Minor Cryptocurrencies

PLOS ONE

Dear Dr. Srivannaboon,

Thank you for submitting your manuscript to PLOS ONE. After careful consideration, we feel that it has merit but does not fully meet PLOS ONE’s publication criteria as it currently stands. Therefore, we invite you to submit a revised version of the manuscript that addresses the points raised during the review process.

At very first glance, the author(s) should emphasize the novelty and originality of the study, as well as the contribution to the related literature. Therewith, several revisions towards introduction, prior literature, as well as quantitative outcomes are necessary.

We look forward to receiving your revised manuscript.

Kind regards,

Stefan Cristian Gherghina, PhD. Habil.

Academic Editor

PLOS ONE

Journal Requirements:

2.We note that you have indicated that data from this study are available upon request. PLOS only allows data to be available upon request if there are legal or ethical restrictions on sharing data publicly. For more information on unacceptable data access restrictions, please see http://journals.plos.org/plosone/s/data-availability#loc-unacceptable-data-access-restrictions.

4. Please include a copy of Table 4 which you refer to in your text on page 10.

Reviewers' comments:

Reviewer's Responses to Questions

**Comments to the Author**

1. Is the manuscript technically sound, and do the data support the conclusions?

Reviewer #1: Partly

Reviewer #2: Yes

Reviewer #3: Yes

2. Has the statistical analysis been performed appropriately and rigorously? 

Reviewer #1: No

Reviewer #2: Yes

Reviewer #3: Yes

3. Have the authors made all data underlying the findings in their manuscript fully available?

Reviewer #1: Yes

Reviewer #2: Yes

Reviewer #3: Yes

4. Is the manuscript presented in an intelligible fashion and written in standard English?

Reviewer #1: Yes

Reviewer #2: Yes

Reviewer #3: Yes

5. Review Comments to the Author

Reviewer #1: The paper confirms empirically the existence of jumps and cojumps for both major and "minor" cryptocurrencies using Andersen and Bollerslev [18] and Barndorff-Nielsen and Shephard [19], as well as well documented statistics for jump analysis.

The existence of jumps has been confirmed by Scaillet et al. [13] for the Bitcoin, and in

Luo, M., Kontosakos, V. E., Pantelous, A. A., & Zhou, J. (2019). Cryptocurrencies: Dust in the wind?. Physica A: Statistical Mechanics and its Applications, 525, 1063-1079,

the authors provided strong statistical evidence to claim that the prices of cryptocurrencies exhibit positive or negative jumps. They also show that the mass-size distribution of aeolian dust particles provide an excellent model. In other words, the generalized hyperbolic (GH) distribution of Barndorff-Nielsen, O. (1977). Exponentially decreasing distributions for the logarithm of particle size. Proceedings of the Royal Society of London. A. Mathematical and Physical Sciences, 353(1674), 401-419, offers a significantly better fit compared to the normal distribution for modelling daily log-returns of cryptocurrencies.

Based on the above, and considering that there are thousands of really minor cryptocurrencies (>2000), such as MERI (~$285k) or BERRY (~$247k) or TEAM (~$107), just to name a few. You just considered OmiseGo, Zcoin, Everex and Average which have market cap >= $40m. Indeed, the chosen cryptocurrencies are by far smaller than Bitcoin, Ethereum (but I would not consider them as minor), the paper does not provide something really novel and interesting. It is just case study among so many others exist in this field.

Reviewer #2: The paper studies jumps and cojumps of cryptocurrencies among them and against SET100. In this sense it is original and the methodology is suitable for the purpose. However, I have some questions and suggestions to the authors:

1. Cryptocurrencies are traded 24 hours a day, whereas Thailand Stock Exchange is only open during certain hours. Therefore, how do you match data from both cryptocurrencies and stock exchange? From a technical point of view, cryptos information flow is continuous, but SET100 will have some periods without trading. Probably the authors should comment on this aspect.

2. The authors should consider some related literature, regarding co-movements. For example:

Eng-Tuck Cheah, Tapas Mishra, Mamata Parhi, Zhuang Zhang, Long Memory Interdependency and Inefficiency in Bitcoin Markets, Economics Letters, Volume 167, 2018,Pages 18-25, https://doi.org/10.1016/j.econlet.2018.02.010.

Paraskevi Katsiampa, Volatility co-movement between Bitcoin and Ether, Finance Research Letters, Volume 30,2019,Pages 221-227,https://doi.org/10.1016/j.frl.2018.10.005.

3. In the empirical findings section, the authors say: "we found that the minor cryptocurrencies have the highest percentage of days with significant jumps". In this aspect,

Bariviera, A. F. (2020). One model is not enough: heterogeneity in cryptocurrencies’ multifractal profiles. Finance Research Letters, (June), 101649. https://doi.org/10.1016/j.frl.2020.101649

already found that small cryptos are prone to sudden jumps. Therefore it could be interesting to link the authors' findings with (probably) the multifractal behavior of small cryptos whose dynamics is more likely to exhibit such jumps.

4. The introduction could be slightly expanded, introducing some discussion on the informational efficiency and portfolio topics. Literature on cryptocurrencies is very abundant. You could select 3 o 4 of the most cited papers in this area. This last recommendation is because the authors discuss (in conclusions) the importance of their findings for regulatory purposes. However they do not comment on the effect that their findings could have for portfolio analysis (portfolios constituted by cryptos only, or cryptos and SET100 stocks).

Reviewer #3: The work analyzes the presence of jumps and co-jumps in cryptocurrencies with high and low market capitalization, using the jump detection structure proposed in Barndorff-Nielsen and Shephard (2004). In general, the methodology is correct, the topic is of interest in the finance and econometrics literatures and the results are important in asset management and risk management. So, I see merits for publication in PLOS ONE, and I have only a few minor comments that I put below.

1 - Page 2 – “However, very few studies have examined the presence of jumps and cojumps in cryptocurrencies.”

Add references on jumps and especially co-jumps in cryptocurrencies. A direct modeling of joint jumps in the mean and conditional variance for cryptocurrencies can be found at Chaim and Laurini (2019), Nonlinear dependence in cryptocurrency markets, The North American Journal of Economics and Finance, Volume 48, 2019, Pages 32-47.

2 - Typo in title of Table 1 – “Jumps and Cojumps Paramter Estimates”

3- Full reference for the cited working papers in the references.

4- Since PLOS ONE is a general public journal, it would be important to post a brief discussion on the importance of jumps and co-jumps in terms of asset allocation and risk management.

6. PLOS authors have the option to publish the peer review history of their article (what does this mean?). If published, this will include your full peer review and any attached files.

Reviewer #1: No

Reviewer #2: No

Reviewer #3: No

---

## [Author Response · Author response to Decision Letter 0]

14 Dec 2020

We would like to take this opportunity to thank the reviewers for their valuable suggestions that have helped to enhance the quality of the paper, entitled “Jumps and Cojumps Analyses of Major and Minor Cryptocurrencies”. We have revised the paper accordingly. Please kindly consider it. Thank you. 

Authors

---

## [Decision Letter · Decision Letter 1]

7 Jan 2021

Jumps and Cojumps Analyses of Major and Minor Cryptocurrencies

PONE-D-20-30502R1

Dear Dr. Srivannaboon,

We’re pleased to inform you that your manuscript has been judged scientifically suitable for publication and will be formally accepted for publication once it meets all outstanding technical requirements. In this regard, the suggestions and recommendations formualted by the first reviewer should be implemented.

Kind regards,

Stefan Cristian Gherghina, PhD. Habil.

Academic Editor

PLOS ONE

Additional Editor Comments (optional):

Reviewers' comments:

Reviewer's Responses to Questions

**Comments to the Author**

1. If the authors have adequately addressed your comments raised in a previous round of review and you feel that this manuscript is now acceptable for publication, you may indicate that here to bypass the “Comments to the Author” section, enter your conflict of interest statement in the “Confidential to Editor” section, and submit your "Accept" recommendation.

Reviewer #1: (No Response)

Reviewer #2: All comments have been addressed

Reviewer #3: All comments have been addressed

2. Is the manuscript technically sound, and do the data support the conclusions?

Reviewer #1: Yes

Reviewer #2: Yes

Reviewer #3: Yes

3. Has the statistical analysis been performed appropriately and rigorously? 

Reviewer #1: Yes

Reviewer #2: Yes

Reviewer #3: Yes

4. Have the authors made all data underlying the findings in their manuscript fully available?

Reviewer #1: Yes

Reviewer #2: No

Reviewer #3: Yes

5. Is the manuscript presented in an intelligible fashion and written in standard English?

Reviewer #1: Yes

Reviewer #2: Yes

Reviewer #3: Yes

6. Review Comments to the Author

Reviewer #1: The paper has been improved significantly. I have two minor points before I could recommend it for publication.

1) I could not identify any discussion in the Introduction about Luo et al. (2019)'s approach and Barndorff-Nielsen (1977) method. I could accept that this might be a point of future research, but I would be very keen to see a brief discussion about how to model the characteristics of jumps and cojumps of cryptocurrencies more precisely in the current version of the paper. It could be also part of the Conclusion as a future research direction.

Personally, I don't think is enough to merely say: "we believe that using this method is beyond the scope of this paper as the method of Barndorff-Nielsen and Shephard (2004) is sufficient to achieve the objective of this paper from the following reasons." It is very subjective, and not scientifically sound as argument.

2) Delete "than" on Table 1 . . . "between than" . . . . 3rd column.

Reviewer #2: (No Response)

Reviewer #3: All my comments have been addressed. In general the revision substantially improved the contribution of the work, and I believe that the work can be published in its current form.

7. PLOS authors have the option to publish the peer review history of their article (what does this mean?). If published, this will include your full peer review and any attached files.

Reviewer #1: No

Reviewer #2: No

Reviewer #3: No

---

## [Editor Report · Acceptance letter]

11 Jan 2021

PONE-D-20-30502R1 

Jumps and Cojumps Analyses of Major and Minor Cryptocurrencies 

Dear Dr. Srivannaboon:

I'm pleased to inform you that your manuscript has been deemed suitable for publication in PLOS ONE. Congratulations! Your manuscript is now with our production department. 

Kind regards, 

on behalf of

Dr. Stefan Cristian Gherghina 

Academic Editor

PLOS ONE